# Multifunctional quantum dot DNA hydrogels

Libing Zhang[1], Sae Rin Jean[2], Sharif Ahmed[1], Peter M. Aldridge[3], Xiyan Li[4], Fengjia Fan[4], Edward H. Sargent[4] & Shana O. Kelley [1,2,3,5]

Biotemplated nanomaterials offer versatile functionality for multimodal imaging, biosensing, and drug delivery. There remains an unmet need for traceable and biocompatible nanomaterials that can be synthesized in a precisely controllable manner. Here, we report self-assembled quantum dot DNA hydrogels that exhibit both size and spectral tunability. We successfully incorporate DNA-templated quantum dots with high quantum yield, long-term photostability, and low cytotoxicity into a hydrogel network in a single step. By leveraging DNA-guided interactions, we introduce multifunctionality for a variety of applications, including enzyme-responsive drug delivery and cell-specific targeting. We report that quantum dot DNA hydrogels can be used for delivery of doxorubicin, an anticancer drug, to increase potency 9-fold against cancer cells. This approach also demonstrated high bio-compatibility, trackability, and in vivo therapeutic efficacy in mice bearing xenografted breast cancer tumors. This work paves the way for the development of new tunable biotemplated nanomaterials with multiple synergistic functionalities for biomedical applications.

---

[1] Department of Pharmaceutical Sciences, Leslie Dan Faculty of Pharmacy, University of Toronto, Toronto, Ontario, Canada M5S 3M2. [2] Department of Chemistry, Faculty of Arts and Science, University of Toronto, Toronto, Ontario, Canada M5S 3H6. [3] Institute for Biomaterials and Biomedical Engineering, University of Toronto, Toronto, Ontario, Canada M5S 3G9. [4] Department of Electrical and Computer Engineering, Faculty of Engineering, University of Toronto, Toronto, Ontario, Canada M5S 3G4. [5] Department of Biochemistry, Faculty of Medicine, University of Toronto, Toronto, Ontario, Canada M5S 1A8. Libing Zhang and Sae Rin Jean authors contributed equally to this work. Correspondence and requests for materials should be addressed to E.H.S. (email: ted.sargent@utoronto.ca) or to S.O.K. (email: shana.kelley@utoronto.ca)

Nanomaterials that can be assembled with a high degree of control over their physical features (e.g., size, shape, or surface charge) are of interest for use in a variety of biological and therapeutic applications. Biotemplated nano-materials provide excellent solubility, low off-target toxicity, and biodegradability[1–5]. However, there is an unmet need for developing trackable nanomaterials (with high quantum yield and photostability) with high biocompatibility that can be readily synthesized in a controllable manner. It would be ideal if these materials could be engineered to include an imaging modality (e.g., a fluorescent tag) that can be traced in living cells without interfering with normal biological functions.

Using DNA as a building block for the assembly of nano-particles is an attractive strategy for the development of hybrid bionanomaterials[6–10]. Through sequence-directed hybridization, DNA molecules have the ability to form predictable two- or even three-dimensional nanostructures[11–16]. DNA hydrogels in particular have attractive features for biological applications such as high solubility, biocompatibility, responsiveness, and versatility[17, 18]. DNA hydrogels are composed of complementary DNA molecules hybridized to form a highly structured network that expands upon hydration in an aqueous environment[12, 13, 19]. These materials can be appended to virtually any type of nucleic acids (e.g., siRNA, miRNA, or aptamer) without the need for chemical ligation and are amenable to loading with DNA-binding drugs[1, 3, 20–22]. DNA-based nanomaterials that can be monitored using fluorescence are highly desirable for biological studies[23–29].

There are several examples of fluorescent hydrogels that have been developed to date such as silver nanocluster DNA hydrogels, quantum dot (QD) polymers or DNA hydrogel/polymer hybrids; however, most require complex multi-step fabrication, have low photostabilities or quantum yields (QY), and do not possess many of the features required for biological studies[11, 17, 30–32]. Additionally, these fluorescent hydrogels have not been designed or fully exploited for biological applications. In this work, we have selected quantum dots as the fluorescent label because of their high photostability and quantum yield, spectral tunability, and ease of incorporation into DNA hydrogels[33–35]. Herein,

we describe the one-step synthesis of self-assembled quantum dot DNA hydrogels (QDHs) with precise control over the size and spectral emission and illustrate their multifunctionality in biological systems.

Previously, QDs have been integrated into hydrogels through chemical conjugation, entrapment, and polymerization[11, 18, 36]. These methods require complex additional steps and reagents and can introduce variability in the composition of the final construct. Here, we leverage DNA complementarity to incorporate DNA-templated QDs into a DNA hydrogel network entirely through self-assembly (Fig. 1a). The fabrication of QDHs was completed in a single step and the gelling process was achieved under physiological conditions. Furthermore, we have shown that we can specifically tune the size of the QDHs to be tailored for specific applications. We show that QDH nanoparticles can enter mammalian cells and exhibit size-dependent endocytic uptake. Interestingly, while QDHs are stable over a range of physiologi-cally relevant temperatures and pH, they are degraded once they enter cells upon nuclease digestion. We leverage this behavior to use the QDHs to deliver a DNA-binding drug doxorubicin (Dox) to cancer cells, and demonstrate that this delivery strategy increases the potency of the drug. Additionally, we functionalized QDHs with aptamers to target specific cell types and also use these DNA-based nanomaterials to modulate protein expression level using siRNA delivery.

## Results

**Synthesis and characterization of QDs.** We adopted a one-pot process that permits the synthesis of QDs using DNA as the template. The DNA ligands are comprised of three parts: a QD-binding domain featuring a phosphorothioate backbone, a spacer, and a DNA-binding domain with phosphate linkages. The phosphorothioate domain has the highest affinity for the cations of the metal chalcogenide QD and thus binds to the surface of the QD, leaving the spacer and DNA-binding domain portion of the sequence unbound[37]. This unbound section of the DNA ligand retains the molecular affinity to complementary DNA sequences. Three distinct colors were observed upon

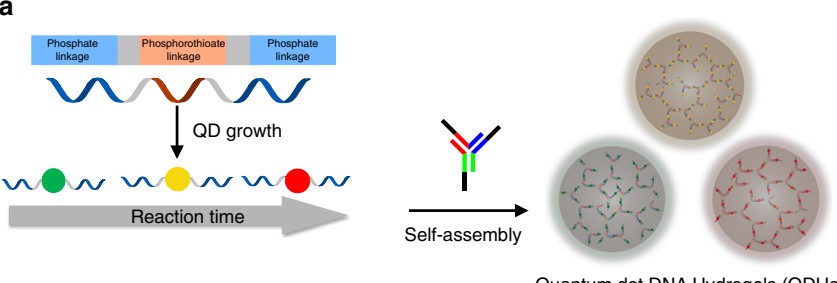

**Fig. 1** Schematic illustrations of synthesis of self-assembled QDHs, modification with therapeutic or targeting agents, and cellular uptake mechanism. **a** Synthesis of DNA-functionalized QDs and the subsequent formation of QDH through hybridization with DNA three-way junction nanostructures. **b** Modifications of QDH for cell-specific targeting with an aptamer and siRNA/drug delivery. Doxorubicin and siRNA are released upon cellular uptake via endocytosis

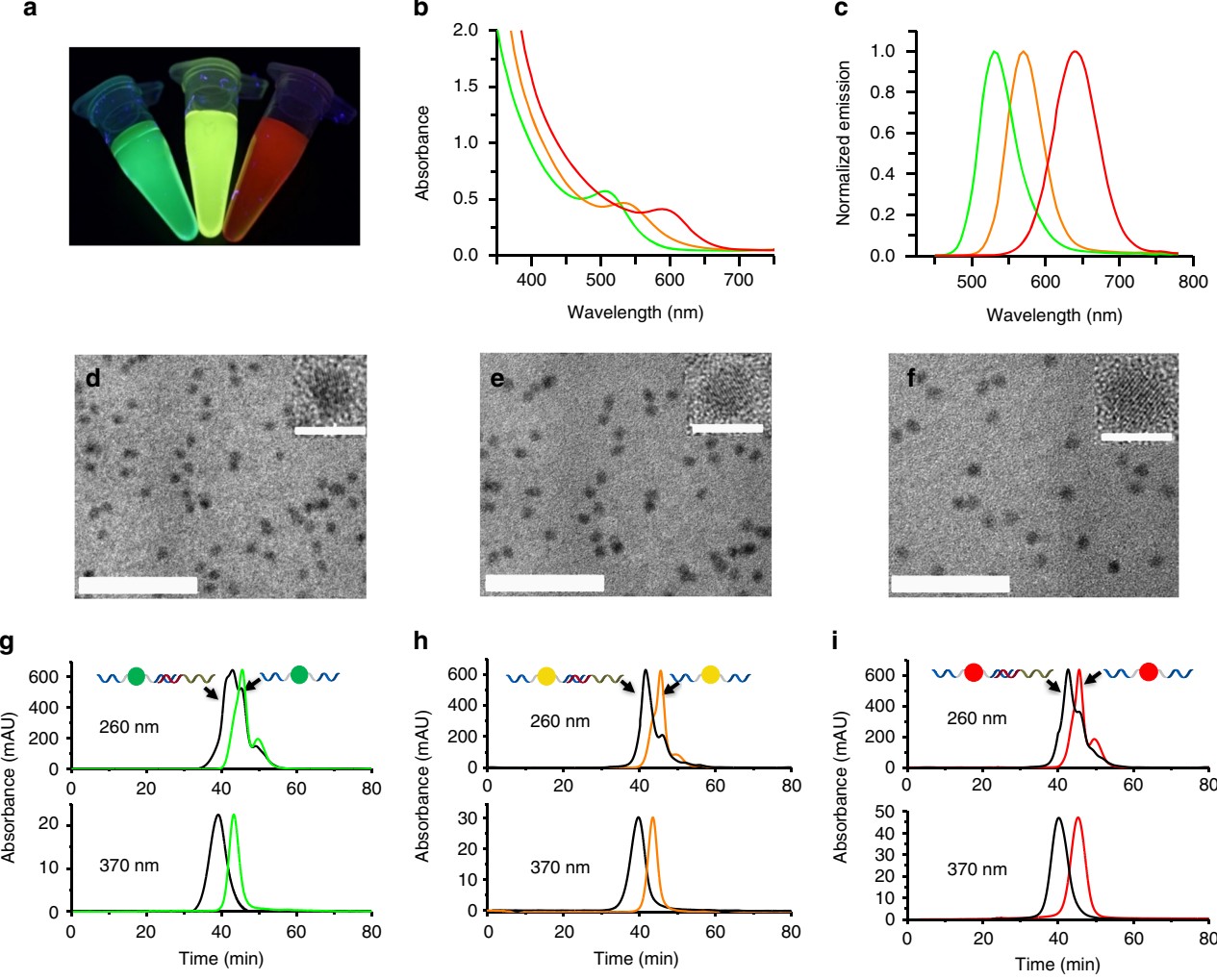

**Fig. 2** Optical, morphological, and hybridization properties of DNA-functionalized QDs. **a** Photograph of the three types of QDs upon illumination with UV light. **b** Absorption spectra of QDs. **c** Normalized fluorescence spectra of QDs. TEMs of the DNA-functionalized green (**d**), yellow (**e**), and red (**f**) CdTe QDs. Scale bars are 50 nm (lower) and 5 nm (upper). **g–i** High-performance liquid chromatography characterization of DNA-functionalized QDs and the hybridization of template with complementary DNA

excitation of the DNA-passivated QDs with UV light (Fig. 2a). For the characterization of spectral properties, UV-Vis and fluorescence emission spectra of DNA-passivated QDs were obtained (Figs. 2b and c). These CdTe QDs exhibit desirable emission properties with high QY values (Supplementary Table 1). In addition, transmission electron microscopy (TEM) was used to analyze the formation of the three DNA-passivated QDs (Figs. 2d–f), which confirms that the materials are nanoscale and monodispersed. Subsequently, high-performance liquid chromatography (HPLC) was used to verify the interaction of DNA with the surface of QDs and the binding ability of phosphate domain with complementary DNA after the DNA has been attached to QDs. The resulting peaks of DNA-passivated QDs at 260 nm (DNA absorbance) and 370 nm (QD absorbance) were recorded (Figs. 2g–i). Compared elution profile of the pure DNA-passivated QDs, a separate peak was observed at 260 nm and the absorption peak at 370 nm shifted in the presence of complementary DNA. Taken together, these results demonstrate that DNA-passivated QDs have been successfully synthesized and the phosphate domain of the DNA template is able to hybridize with the complementary target DNA.

In order to determine whether we could achieve self-assembly of the designed QD/DNA and a DNA-three way junction (TWJ)

nanostructure, native polyacrylamide gel electrophoresis (PAGE) was performed to verify the stepwise assembly of DNA nanostructures (Fig. 3a). As each DNA strand was added, we observed a distinct band shift due to the increase in weight of the DNA complex, indicating the effectiveness of DNA assembly. By comparing the bands in these lanes, we concluded that the QD/DNA complexes and DNA-TWJ nanostructures hybridize together to form crosslinked DNA nanostructures with a distinct band corresponding to a higher molecular weight. Next, we sought to determine if the DNA-functionalized QD and DNA-TWJ nanostructure can be combined to form DNA hydrogels. The ideal molar ratio of the DNA-TWJ nanostructure and QD/DNA was calculated to be 1:1.5, and the ratio of their sticky ends was 1:1. This ratio ensures that all of the starting materials are consumed. Following the self-assembly of these materials, we observed that the solution lost its fluidity and appeared to be gel-like, suggesting that a sol-to-gel transition has occurred (Fig. 3b). Under the irradiation of UV light, we observed the corresponding fluorescence of the QDHs. It is important to note that these QDHs demonstrated high photostability over period of one week (Supplementary Fig. 2a). Furthermore, rheological characterization was carried out to confirm the formation of the DNA hydrogel (Fig. 3c). The shear-storage modulus (G′) was

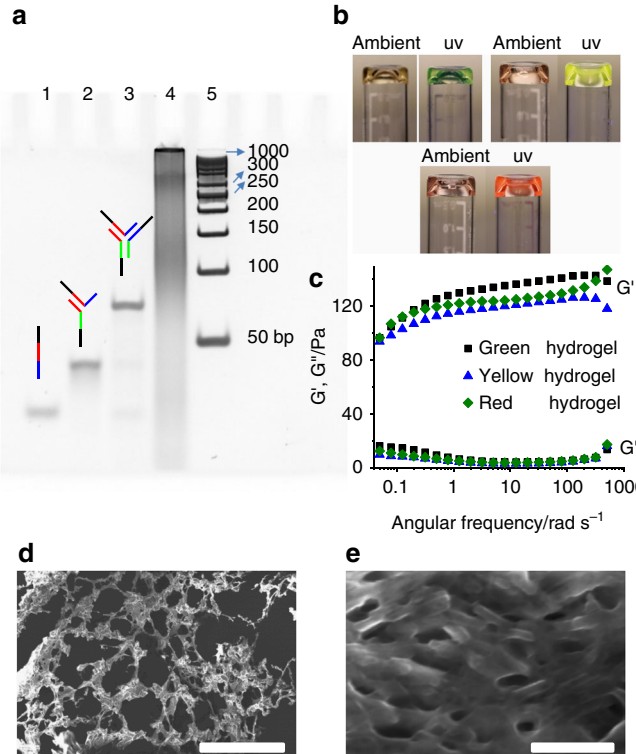

**Fig. 3** Self-assembly and hydrogel formation of QDHs. **a** Native PAGE analysis to verify self-assembly of the QD-DNA and Y-shaped DNA nanostructures: Lane 1: Y1; lane 2: Y1 + Y2; lane 3: Y1 + Y2 + Y3; lane 4: DNA-TWJ + QD-DNA; lane 5: DNA ladder. **b** Photographs of QDHs under irradiation with ambient light and ultraviolet light. **c** Frequency sweep test were carried out between 0.05 and 500 rad/s at a fixed strain of 5% for the three color hydrogels. SEM images of DNA hydrogels with low resolution (**d**) scale bar: 500 μm and high resolution (**e**) scale bar: 500 nm

constantly higher than the shear-loss modulus (G″) over the entire frequency range, providing a clear signature of the gel-like state. In addition, the G′ showed a slight frequency dependent increase, suggesting that a physical gel was produced. Lastly, scanning electron microscopy (SEM) images of the DNA hydrogel structure were obtained (Figs. 3d and e). In the dry state, a porous crosslinked structure of the hydrogel was observed. From these results, we can conclude that QDHs are indeed formed by the self-assembly of QD/DNA and DNA-TWJ nanostructure with specific characteristics of a hydrogel.

We found that the size and the degree of swelling of QDHs are controlled by the initial concentration. The higher the initial concentration, the higher the degree of swelling of the hydrogel. More specifically, the QDH swelled >500% when combined with 0.2 mM DNA-passivated QD (Supplementary Table 2). Dynamic light scattering (DLS) was used to monitor the size of QDHs synthesized with different initial concentrations (Supplementary Fig. 1a–f). We found that the diameter of QDHs increased as we increased the initial concentration of DNA-functionalized QD in a precise and predictable manner (Supplementary Fig. 2b). Zeta potential of the QDH was also investigated (Supplementary Fig. 2c. The zeta potential measurements for QDH showed that particles had a net negative charge of −22.3 ± 1.54 mV.

In order to study the structural durability or responsiveness of QDHs under physiologically relevant conditions, we investigated the stability of QDHs in a variety of environments. First, we studied the enzymatic responsiveness of the designed DNA hydrogels. DNase I is a robust enzyme that nonspecifically cleaves single and double-stranded DNA. When the QDH was exposed to DNaseI the network composed of DNA was degraded and the DNA hydrogel was converted to the sol state (Supplementary Fig. 2d). Subsequently, we evaluated the stability at different biologically relevant pH and in cell media that was used for the biological assays. We found that the hydrogel is stable in pH range 5.0 to 9.0 (Supplementary Fig. 2e). In cell media, we incubated the hydrogel for 3, 24, and 72 h at 37 °C to test its stability (Supplementary Fig. 2f). We found the size increased slightly with the incubation time in MEM-α medium, which is likely due to the hybridization of the hydrogel with free residual DNA in the solution. While in conditioned cell medium and 10% serum, the QDH demonstrated high stability with only < 15% reduction in diameter after 72 h incubation.

**Cellular uptake and toxicity of QDHs.** We sought to investigate if the QDHs with varying diameter (80, 100, and 140 nm) could permeate cells. Indeed, when we imaged the QDHs using confocal microscopy, all three sizes showed strong intracellular localization (Fig. 4a). As illustrated by the lack of overlap with a nuclear dye Hoechst in blue, the QDHs exhibited a primarily extranuclear localization. Next, we studied the effect of the size of QDHs on the cellular uptake using flow cytometry (Fig. 4b). Interestingly, we found that decreasing the size of QDHs resulted in an increase in cellular uptake. Additionally, all sizes of QDH had higher cellular uptake than QD alone. Owing to its high cell permeability, we selected the smallest QDH with 80 nm diameter for further studies. Using this QDH, we elucidated the mechanism of its uptake in depth. Based on the size of the QDH, we evaluated the various endocytic pathways as potential uptake mechanisms. To this end, we pretreated the cells with inhibitors of distinct endocytic pathways with varying diameter of vesicles: dynasore (clathrin-mediated endocytosis, ~120 nm), filipin III (caveolin-mediated endocytosis, ~ 60 nm), and cytochalasin D (macropinocytosis, > 1 μm). Following treatment with QDH, we analyzed the effect of these inhibitors on the cellular uptake (Fig. 4c). As expected, only the pathways with vesicles with > 80 nm diameter that could enclose the QDHs were affected (clathrin-mediated endocytosis and macropinocytosis).

In order for QDHs to be useful in biological applications, they must exhibit minimal cellular toxicity. Consistent with previous reports on toxicity of QDs and other metal complexes, our initial QDHs exhibited significant toxicity in HeLa cells whereas the DNA hydrogel itself was non-toxic (Fig. 4d). We therefore worked to modify the QDs included in the QDH to make them inert to the cellular environment[38]. One approach is to coat the QDs with ZnS shell to reduce the $Cd^{2+}$ leaching into the cells. When we applied this strategy to the QDH to generate ZnS-QDH, we observed a significantly reduced cellular toxicity. ZnS-QDHs were synthesized and characterized similarly to QDH (Supplementary Figs. 3a–f). DNA hydrogel alone displayed negligible toxicity, supplanted by the finding that showed QDs and ZnS-QDs had similar toxicity to their DNA hydrogel constructs (Supplementary Fig. 5a). Additionally, we observed a significantly higher QY by using CdSe as the QD core compared to CdTe (Supplementary Table 1). We moved forward with the optimized ZnS-QDH construct for studying the potential use of QDHs in biological or therapeutic applications.

**Cellular delivery applications of QDHs.** The first application we explored was the potential of QDHs to be used for cell-specific targeting and siRNA delivery. Ideally, targeted drug delivery vehicles would deliver cargo specifically to cancer cells. We appended a DNA-based aptamer that is specific to a cell-surface receptor on CCL-119 cells to DNA sequences on the surface of

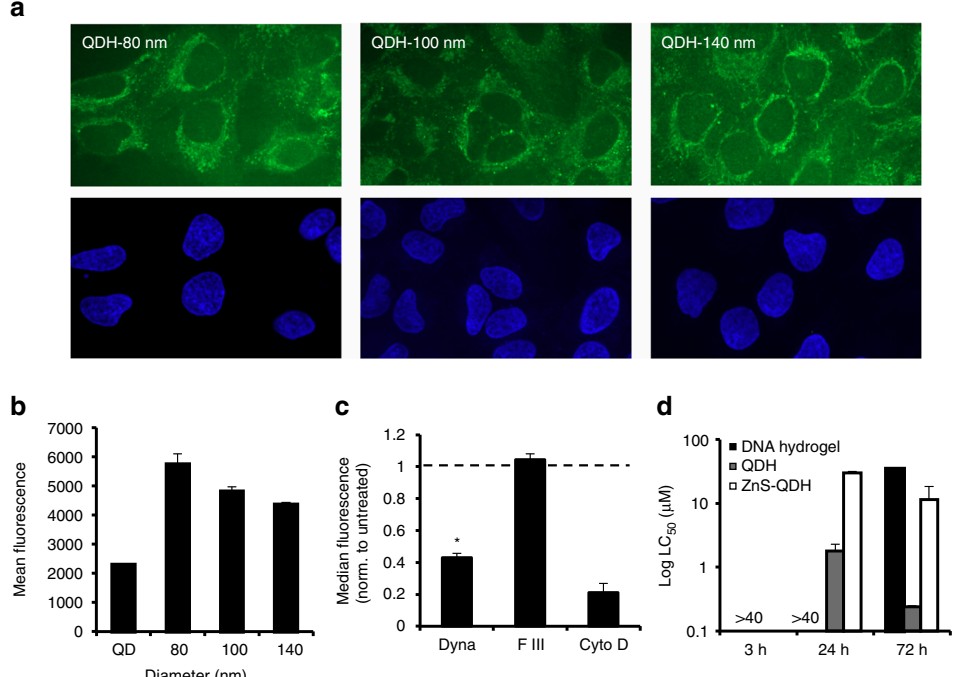

**Fig. 4** Uptake and viability of QDHs in cells. **a** Cellular accumulation of 80 nm, 100 nm, and 140 nm QDH (*green*) excluded from nuclei (*blue*) of HeLa cells. **b** Quantification of cellular uptake of QD alone and QDH (80 nm, 100 nm, and 140 nm) in HeLa cells by flow cytometry ($n = 3$). **c** Elucidation of endocytic uptake method of 80 nm QDH in HeLa cells by flow cytometry ($n = 4$). Dyna: dynasore,; F III: filipin III; Cyto D: cytochalasin D. One-way ANOVA for treated vs. untreated control *$P < 0.01$. **d** Cell viability in HeLa cells. Comparison of $LC_{50}$ values of DNA hydrogel, QDH, and ZnS-QDH at 3, 24, and 72 h time points ($n = 3$). All error bars represent s.e.m

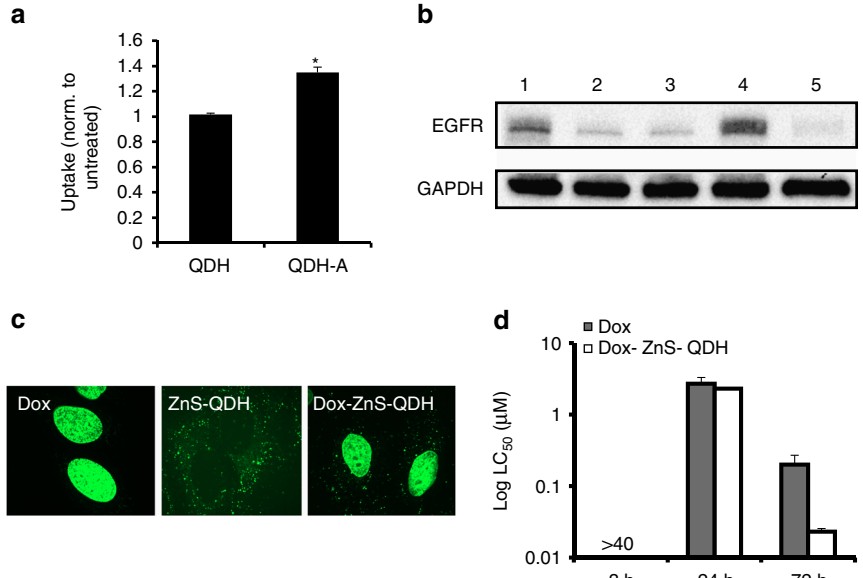

**Fig. 5** Cell-specific targeting, siRNA transfection, and Dox delivery using QDHs. **a** Cell-specific targeting with aptamer in CCL-119 cells. Uptake of QDH and QDH-A were measured using flow cytometry. One-way ANOVA, *$P < 0.05$. **b** Determination of siRNA delivery with QDH. EGFR was knocked down in HeLa cells with: lane 1: Untreated; 2: EGFR siRNA; 3: EGFR siRNA-aptamer; 4: silencer; 5: QDH-EGFR siRNA. **c** Cellular localization of Dox, ZnS-QDH, and Dox-ZnS-QDH. **d** Comparison of $LC_{50}$ values of Dox and Dox-ZnS-QDH at 3, 24, and 72 h. All error bars represent s.e.m. ($n = 4$). See Supplementary Fig. 8 for full blot

the QDH through a simple hybridization step to generate QDH-aptamer (Supplementary Fig. 4a). By employing this cell-specific targeting approach, we significantly increased the uptake of QDHs to CCL-119 leukemic cells (Fig. 5a)[39]. In contrast, in Ramos cells that do not express the cell-surface receptor, QDH-aptamer had similar cellular uptake as QDH (Supplementary

Fig. 5b). Another use of the unhybridized DNA sequences on the surface of QDHs is to hybridize it to nucleic acids for delivery into cells. Gene or siRNA delivery is a powerful method by which nucleic acids are targeted intracellularly in order to introduce new proteins or modulate protein expression levels for cellular studies or therapeutic applications. As a proof-of-concept, we attached a

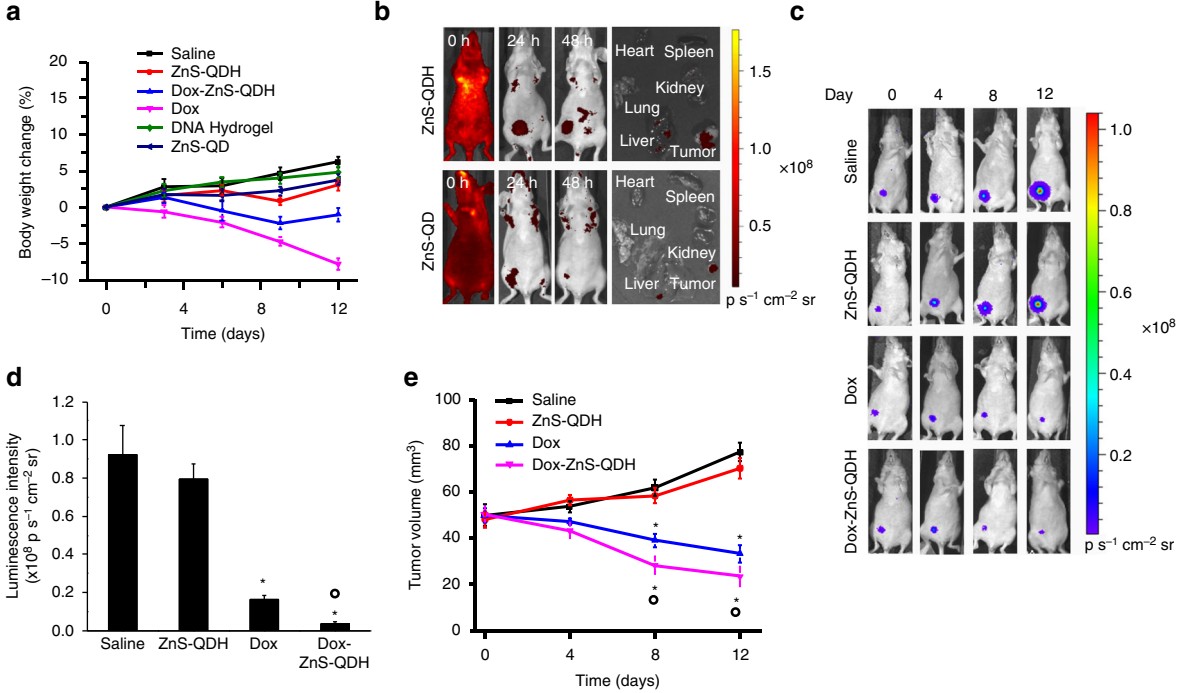

**Fig. 6** Analysis of bioavailability and in vivo efficacy. **a** Maximum tolerable dose of saline, ZnS-QDH, Dox, Dox-ZnS-QDH, DNA hydrogel, and ZnS-QD treated healthy mice measured by percentage of body weight change from day 0 to 12 ($n = 5$ per group). **b** Biodistribution of ZnS-QD and ZnS-QDH after 48 h treatment. **c** Bioluminescence images of tumors of treated mice from day 0 to 12. **d** Quantification of bioluminescence on day 12 of treatment regimen. **e** Analysis of tumor volume of treated MDA-MB-231 breast cancer xenograft mice ($n = 5$ per group). All error bars represent s.e.m. ($n = 5$). ANOVA one-way analysis of treated vs. saline control *$P < 0.05$, vs. Dox °$P < 0.05$

siRNA to reduce the expression of epidermal growth factor receptor (EGFR) to QDH using RNAi technology (Supplementary Fig. 4b). We confirmed that QDH-siRNA accumulates in cells effectively (Supplementary Fig. 4c). Interestingly, QDH-siRNA induced a significant reduction in EGFR expression compared to untreated and to a similar level with EGFR siRNA alone (Fig. 5b). This shows that we are able to modulate protein levels using QDH as a delivery vector, eliminating the need for using toxic transfection agents. In addition, this strategy can be combined with drug delivery and cell-specific targeting to synergistically enhance the therapeutic efficacy.

We tested the ability of QDH to be used as a delivery vector for small molecule drugs. To this end, we selected a potent anticancer drug doxorubicin (Dox) since it preferentially intercalates into DNA for facile loading. Since the innate fluorescence of Dox is stoichiometrically quenched by DNA (Supplementary Fig. 4d), the Dox content in QDH is easily quantifiable. We compared the cellular localization of Dox, ZnS-QDH, and Dox-ZnS-QDH (Fig. 5c). In contrast to the extranuclear staining of ZnS-QDH, we observed a high nuclear localization with Dox-ZnS-QDH similar to Dox, indicating that we were able to successfully release Dox intracellularly. Additionally, we confirmed that Dox is not prematurely released from the DNA hydrogel in cell media or PBS up to 72 h at 37 °C and is only released in the presence of DNase (Supplementary Fig. 4e). This suggests that we are able to specifically release Dox inside cells. Interestingly, when we compared the $LC_{50}$ values of Dox-ZnS-QDH to the parent compound Dox, we found that at 72 h treatment period, Dox-ZnS-QDH was ninefold more effective at killing cancer cells (Fig. 5d). We posited that this may be due to the controlled release of Dox in the vicinity of the site of action that permits higher retention in cells. Compared to related constructs such as Dox-dsDNA (Dox-loaded linear dsDNA) or Dox-YDNA (Dox-loaded Y-shaped DNA nanostructures that do not form a

hydrogel network), Dox-DH demonstrated higher cellular uptake and potency (Supplementary Fig. 5c, d). These findings suggest that DNA hydrogel constructs are ideal vectors for the delivery of DNA-damaging agents such as Dox in cancer cells.

**Testing of QDHs in xenograft model.** We sought to test whether ZnS-QDH could be used as drug carriers in an animal model. The toxicity in mice treated with saline, QD, DNA hydrogel, Dox, ZnS-QDH, and Dox-ZnS-QDH was evaluated by determining the maximum tolerated dose (MTD; Fig. 6a). The samples were intravenously injected into healthy mice every 3 days up to 12 days and body weight was documented throughout the treatment period. All treated mice maintained less than 10% weight loss, and no changes in behavior upon injection were observed. Major organs (heart, liver, and kidney) were collected from all groups from MTD analysis for histological analysis by haematoxylin and eosin (H&E) staining. As shown in Supplementary Fig. 6a–c, only a single small randomly distributed necroinflammatory foci were found in the liver of one of the Dox treated mice (indicated with arrow). No other toxicologically significant lesions were found in any other organ. Next, biodistribution of ZnS-QDH was studied in the xenograft model. ZnS-QDH and ZnS-QD were administered by tail-vein injection and monitored for fluorescence up to 48 h. Mice were sacrificed 48 h post injection and the major organs were collected for fluorescence imaging to study biodistribution (Fig. 6b). ZnS-QDH and ZnS-QD mainly accumulated in tumors potentially due to the EPR (enhanced permeability and retention) effect, which is consistent with the biodistribution of DNA origami in vivo[40]. As expected, fluorescence signal was also observed in the liver to a lesser extent, however other organs were largely unexposed. Quantification of the fluorescence intensity of major organs revealed that ZnS-QDH and ZnS-QD had similar levels of

uptake with highest accumulation in the liver. Interestingly, ZnS-QDH accumulated 2.8-fold higher than ZnS-QD in the tumor (Supplementary Fig. 7). High tumor accumulation can be used to increase the clinical utility of drugs such as Dox that often result in off-target effects such as cardiotoxicity. Taken together, ZnS-QDH possesses high biocompatibility and provides an excellent platform for bioimaging and drug delivery in disease-relevant in vivo systems. Thus, the dosing regimen used for MTD was applied to the efficacy analysis in breast cancer xenograft mice.

To assess the therapeutic potential of ZnS-QDH as drug carriers, the sizes of the tumors of treated mice were monitored. MDA-MA-231/Luc breast cancer cells were implanted subcutaneously into mice until tumor size reached approximately 50 mm$^3$. The tumor-bearing mice were randomly divided into 4 groups and the samples were administered into the mice every 3 days up to 12 days by tail vein injection. During the treatment period, bioluminescence images and caliper measurements were recorded to assess tumor growth, and after the completion of the study H&E staining was conducted (Supplementary Fig. 6). The mice bearing tumors derived from MDA-MB-231/Luc breast cancer cells display bioluminescence signal, which is directly related to the size of the tumors. The bioluminescence intensity decreased over the treatment period only in mice treated with Dox or Dox-ZnS-QDH compared to the saline control (Fig. 6c). Quantification of the bioluminescence intensity of treated mice on day 12 of treatment revealed that Dox-ZnS-QDH reduced the tumor size by 20-fold (Fig. 6d). This result confirmed that Dox-ZnS-QDH is significantly more efficacious than Dox (threefold), the frontline drug used clinically for cancer treatment. Mice treated with Dox and Dox-ZnS-QDH both significantly reduced tumor volume (62%) compared to saline treated mice (Fig. 6e). Interestingly, Dox-ZnS-QDH treatment displayed significantly higher reduction in tumor volume than the Dox treated group, likely owing to higher uptake from the EPR effect. As shown in Figs. 6c–e, ZnS-QDH treated mice bearing did not reduce the tumor volume or size significantly, indicating that the high efficacy of Dox-ZnS-QDH does not originate from non-specific toxicity from the carrier.

This work showcases the versatility of traceable and biocompatible QDHs for various biomedical applications, leveraging the DNA-specific interactions and biocompatibility. We demonstrated that QDHs can act as excellent nanocarriers for cell-specific targeting and drug delivery. More broadly, these biotemplated nanomaterials show immense potential in multiplexed imaging studies and rationally-designed synergistic biomedical functionality. For future directions, implementing biosensing technology and spatiotemporal control of release of biologically active compounds in cellular or animal models should be explored.

## Methods

**Materials**. Cadmium chloride (CaCl$_2$, > 99.99%), tellurium powder (Te, > 99.8%), sodium borohydride (NaBH$_4$, > 98%), 3-Mercaptopropionic acid (MPA, > 99.0%), cadmium oxide (CdO, > 99.99%), zinc acetate dihydrate (Zn(AC)$_2$· 2H$_2$O, 99.99%), sulfur powder (S, > 99.5%), selenium powder (Se, > 99.99%), oleylamine (OLA, 80–90%), octadecene (ODE, 90%), oleic acid (OA, 90%), tri-octylphosphine (TOP, 90%) and 1-octanethiol (> 98.5%) were purchased from Sigma-Aldrich. All DNA and RNA were purchased from Integrated DNA Technologies (see Supplementary Table 3 for sequences).

**Cell culture**. HeLa, Ramos, and CCL-119 cells were obtained from the American Type Culture Collection (ATCC). HeLa cells were cultured in MEM-α (Thermo) supplemented with 10% (v/v) fetal bovine serum (FBS, Thermo). CCL-119 and Ramos cells were cultured in RPMI 1640 (Thermo) supplemented with 10% (v/v) FBS. MDA-MB-231/Luc human breast cancer cell line purchased from Cell Biolabs Inc. was grown in DMEM (High Glucose) supplemented with 10% FBS, 0.1 mM MEM Non-Essential Amino Acids (NEAA) and 2 mM L-glutamine. All cells were grown in T75 cm$^2$ flasks with vented caps (Sarstedt) in a humidified incubator at

37 °C with 5% CO$_2$. STR (short tandem repeat) profiling and mycoplasma test were done for all cell lines in culture. No contamination was found in any of the cell lines used for experiments.

**Synthesis of DNA-capped QDs**. Synthesis of DNA-capped CdTe QDs: sodium hydrogen telluride (NaHTe) was prepared by dissolving 0.025 g sodium borohydride and 0.040 g tellurium powder in 1 mL water and sonicating for 30 min; the reaction was vented using a needle inserted through the lid of the reaction vessel. For the synthesis of CdTe QDs, 10 µl cadmium chloride (100 mM) and 2 µL MPA precursor (500 mM) were added into water to a total volume of 1 ml. pH was adjusted to 9.0 with 1.0 M NaOH solution. After adding 2.0 µL Te precursor to 100 nmol chimeric DNA solution, the solution was heated to 100 °C for 30 min, 1.0 h, and 4.0 h to obtain green, yellow and red QDs, respectively. DNA-capped ZnS-QDs: 2.98 g CdO was dissolved in 40 mL oleic acid at 170 °C under vacuum. 2.45 g Zn(AC)$_2$·2H$_2$O was dissolved in OLA at 170 °C under vacuum until a clear light pink solution was obtained. CdSe QDs were synthesized using existing literature protocol[41]. Two hundred forty milligrams CdO, 24 g TOPO and 1.12 g ODPA were mixed in a 100 ml three-neck flask, the mixture was heated to 150 °C for 0.5 h under vacuum, then kept at 320 °C for 2 h under nitrogen. 4 mL of TOP were injected into the flask and incubated at 380 °C. The injected selenium precursor consisted of 2 ml selenium in TOP solution at a concentration of 60 mg ml$^{-1}$. The reaction was terminated adding acetone. For shell growth, the nanoparticles were redispersed in hexane. The synthesis of CdSe core and CdSe/CdS/ZnS core-shell-shell structures was adapted from a previously described procedure[42]. For shell growth, a CdSe QD dispersion was added into a mixture of 12 ml OLA and 12 ml ODE, and pumped in vacuum at 100 °C for approximately 0.5 h; the reaction was then exposed to N$_2$. 3 mL of Cd-oleate was diluted in 21 ml ODE and 320 µl octanethiol was diluted in 24 ml ODE. Cd-oleate and octanethiol solutions were injected simultaneously at a rate of 12 ml h$^{-1}$ while the temperature was ramped from 100 to 310 °C. After CdS shell growth, the solution was cooled to 290 °C. 1.5 ml of prepared Zn-OLA diluted in 10.5 ml ODE and 0.03 g sulfur dissolved in 2 ml OLA was mixed and continuously injected at a speed of 14 ml h$^{-1}$ at 290 °C to grow the ZnS shell. The solution was annealed for 10 min at 290 °C; then 4 ml OA was added followed by further annealling at 290 °C for 10 min. The final products were precipitated using acetone, and re-dispersed in hexane. The ZnS-QDs and DNA template were mixed in aqueous solution and incubated for 2 h at room temperature; the products were purified with Microsep™ Centrifugal Devices (YM-100, Pall Corporation).

**Optical characterization**. UV-Vis absorbance spectra and fluorescence emission spectra of QDs were recorded using a SpectraMax M2 (Molecular Devices Corporation). The fluorescence emission spectra of Dox were also recorded using a SpectraMax M2 using excitation at 480 nm.

**TEM and SEM characterization**. Transmission electron microscopy (TEM) measurements were performed using Hitachi H-8100 EM with an accelerating voltage of 300 kV. Scanning electron microscopy (SEM) was performed using a Quanta FEG 250 (10 kV).

**Quantum yield calculation**. The QY was calculated using a standard method[43], which was performed using a Quanta-Phi integrating sphere coupled to a Horiba Fluorolog system. A monochromated Xe lamp was used as an excitation source. Parameters used in the measurement are as follows: excitation wavelength = 440 nm; bandpass values of 2 and 2 nm for the excitation and emission slits; step increments = 1 nm; integration time = 0.1 s per data point. Excitation and emission spectra were collected with the sample directly in the excitation beam path and with a sample offset from the beam path and the empty sphere. A neutral density filter with a known transmission was used to measure the excitation intensity.

**High-performance liquid chromatography**. High performance liquid chromatography (HPLC) was performed with a Superose 6 10/300 GL column (GE Healthcare) conjugated to an Agilent 1200 Infinite HPLC system. Each sample was centrifuged at 10000 rpm for 5 min to remove any insoluble aggregates before injection. The flow rate was 0.35 ml min$^{-1}$, the injection volume was ~ 80 µl and each sample was run for an overall time of 1.3 h. The absorption wavelength used to monitor DNA was 260 nm and QDs was 370 nm. The running buffer used was PBS, pH 7.

**Dynamic light scattering**. Hydrodynamic sizes and zeta potential were measured using Zetasizer Nano ZS90 (Malvern Instruments Ltd.). The conditions of these experiments were identical to the HPLC trials described above.

**Gel electrophoresis**. For native polyacrylamide gel electrophoresis (PAGE), the DNA solution was mixed with loading buffer and run on a 10% native poly-acrylamide gel. The electrophoresis was conducted in 1 × TAE buffer at constant voltage of 110 V for 1 h. The gels were scanned by a UV transilluminator after staining with SYBR green. For agarose gel electrophoresis, the QD samples were

mixed with loading buffer and run on 0.5% agarose gel electrophoresis for 20 min at 100 V in TAE buffer. The gel was imaged using a standard ultraviolet gel box.

**Assembly of QDH**. DNA-capped QDs were firstly purified with MicrosepTM Advance Centrifugal Devices via centrifugation at 10,000 rpm for 3 min to remove inorganic salts and excess DNA. The purification was repeated twice. The purified QDs were then recovered in Tris-HCl buffer (50 mM, 12.5 mM $Mg^{2+}$, pH = 8.5) for assembly. DNA-capped QDs and Y-shaped DNA nanostructure (ratio: 1.5:1) were mixed and incubated at 37 °C for 2 h. The QDHs were subsequently purified with MicrosepTM Advance Centrifugal Devices via centrifugation at 10,000 rpm for 3 min to remove free QDs and DNA. The purified fraction was resuspended in PBS buffer.

**Rheological tests**. Rheological tests were carried out on an AR2000 Rheometer (TA Instruments). Frequency sweep tests were carried out on mixtures between 0.05 and 500 rad/s at 25 °C at a fixed strain of 5% using a 40 mm, 0.5° cone.

**Preparation of QDH-aptamer**. The QDH was incubated with cell binding aptamer (4.0 µM) in 1 × PBS buffer (12.5 mM $Mg^{2+}$). The hybridization reaction was carried out at room temperature for 1 h. Functionalized QDH were buffer exchanged using a Microcon YM-100 centrifugal filter by resuspending the QDH in cell binding buffer.

**Preparation of QDH-siRNA**. The QDH was incubated with mis-cy5 (4.0 µM) in 1 × PBS buffer (12.5 mM $Mg^{2+}$) for 1 h and then purified with MicrosepTM Advance Centrifugal Devices via centrifugation at 10,000 rpm for 3 min to remove free mis-cy5. And the free mis-cy5 quantified by measuring the fluorescence of cy5 at 668 nm. The siRNA with exact complementary sequence was added into the QDH/mis-cy5 solution to replace mis-cy5 through strand displacement reaction and then purified with MicrosepTM Advance Centrifugal Devices to obtain QDH-siRNA (Supplementary Fig. 4d).

**Dox loading**. Doxorubicin solution (2 mM) was incubated with the QDH for 1 h and then purified with MicrosepTM Advance Centrifugal Devices via centrifugation at 10,000 rpm for 5 min to remove free doxorubicin. And the free doxorubicin quantified by measuring the absorption of Dox at 480 nm. The Dox loading content and efficiency of loading into the QDHs were indirectly calculated by Dox loading content at the beginning subtract the free doxorubicin content. The purified fraction was resuspended in PBS buffer.

**Cell viability**. HeLa cells were seeded in 96-well plates (Sarstedt) at 4000 cells per well for 3 or 24 h treatments and 1000 cells per well for 72 h treatment 1 day prior. Cells were treated with appropriate cell media containing 10% FBS. Cell viability was measured after treatment with Cell Counting Kit-8 (Donjindo Molecular Technologies) as per manufacturer's instructions.

**Cellular uptake by flow cytometry**. HeLa cells were seeded in 12-well plates (Sarstedt) at 20,000 cells per well 1 day prior to experiment. For cell-specific targeting experiments, 60,000 cells of CCL-119 cells were used per sample. Cells were treated with 12 µM of QD, QDH-80, QDH-100, and QDH-140 or 48 µM QDH and QDH-aptamer at 37 °C for 3 h. For the endocytic uptake pathway determination, cells were pretreated with the following endocytosis inhibitors 1 h prior to QDH treatment: 200 µM dynasore, 4 µM filipin III, or 400 µM cytochalasin D (Santa Cruz Biotechnology). After treatment, cells were washed once with media then harvested using 0.25% Trypsin-EDTA (Thermo) and centrifuged at $500 \times g$ for 5 mins. The cell pellets were resuspended in 1 ml PBS and centrifuged again. To each sample, 400 µl of PBS containing 5 nM Sytox Red (Thermo) was added for 10 mins at room temperature. The samples were vortexed and analyzed on BD FACSCanto flow cytometer (BD Biosciences) using 488 nm laser excitation for quantification of QDH uptake. The cell-specific targeting measurements were performed as described previously[39].

**Confocal microscopy**. HeLa cells were seeded at 8000 cells per well in eight-well µ-slides (Ibidi) 1 day prior to experiment. Cells were treated with 3 µM QDHs in MEM-α with 10% FBS (v/v) for 3 h at 37 °C. A final concentration of 50 nM Hoechst dye (Thermo) was added in the last 30 mins of treatment. Cells were washed three times using MEM-α without FBS and once with PBS then fixed using 4% PFA in PBS for 10 mins. The plate was washed twice with PBS and mounted with mounting medium (Ibidi). The imaging was performed on Zeiss spinning disk confocal microscope. The nuclear regions of the cells were used to measure Pearson's correlation coefficient values with Zen software.

**siRNA transfection and protein quantification**l. HeLa cells were seeded at 40,000 cells per T75 cm² flask 1 day prior to transfection. HeLa cells were treated with siRNA using Lipofectamine 2000 (Thermo) as per manufacturer's instructions or with QDH-siRNA for 72 h at 37 °C with 5% CO₂. Following treatment, cells were harvested using 0.25% Trypsin-EDTA. Cells were washed twice with ice-cold PBS

and lysed using RIPA buffer containing PMSF and Protease/Phosphatase Inhibitor Cocktail (Cell Signaling Technology) for 30 mins on ice with occasional vortexing. Cell lysate was centrifuged at $16,000 \times g$ for 5 mins at 4 °C and the supernatant was collected and quantified for protein concentration with 660 nm Protein Assay (Thermo) as per manufacturer's instructions.

**Western blot analysis**. The protein extracts (20–30 µg) were loaded on a 4–15% gradient TGX precast protein gel (Bio-Rad). The resulting gel was transferred to a PVDF membrane and blocked in 5% BSA in TBST buffer for 1 h. Membrane was probed overnight at 4 °C or for 1 h at room temperature with primary antibodies as follows: 1:200 EGFR (ab2430, Abcam) or 1:1000 GAPDH (2118 S, New England Biolabs) in 2.5% BSA in TBST. The membrane was rinsed 3 times with TBST for 5 min each and immunoblotted with a secondary antibody, 1:1000 anti-rabbit IgG-HRP linked antibody (7074 S, Cell Signaling Technology) for 1 h in 2.5% BSA in TBST. The blot was rinsed three times with TBST for 5 min each and ECL Western Blotting Substrate (Thermo) was used to detect bands on the VersaDoc Imager (Bio-Rad).

**MDA-MB-231/Luc orthotropic tumor xenograft Model**. All animal experiments were carried out in accordance with the protocol approved by the University Health Network (UHN) Animal Care Committee. 6 to 8-week-old female mice (Athymic Nude-$Foxn1^{nu}$, Envigo) were maintained at the UHN animal facility. The orthotropic tumor model was established by injecting 50 µl of MDA-MB-231/Luc cell suspension ($1 \times 10^6$ cells per ml) into the inguinal mammary fat pad of athymic nude mice.

**Sample dosage**. The dosage in Dox-ZnS-QDH and Dox treatment of 4 mg kg$^{-1}$ was determined using equimolar amount of Dox. The amount of ZnS-QDH (0.45 mg kg$^{-1}$) used was based on the same weight of carrier used in Dox-ZnS-QDH group. The dosage of DNA hydrogel and ZnS-QD were determined to be 0.45 mg kg$^{-1}$. The control group received an equal amount of 0.9% saline injection. The samples were administered into mice through tail-vein injection by an animal technician in a blinded sample injection.

**Mouse body weight and histological analysis**. The mice ($n = 30$) used in each experiment were randomly divided into 6 groups ($n = 5$ per group). The weight of the healthy mice used for MTD was measured using an electronic balance every 3 days up to 12 days. Subsequently, mice were euthanized and heart, liver, and kidney were extracted and fixed in 10% buffered formalin. Tissue sections were stained with hematoxylin and eosin and blinded histological examination was performed by a veterinary pathologist.

**Tumor volume measurements**. For efficacy testing, treatment was administered via tail vein injection every 3 days up to 12 days once the tumor size reached ~50 mm³. The tumor volume was measured by using a digital caliper and calculated using the formula $V = (L \times W^2)/2$, where $L$ and $W$ are the length and width of the tumor, respectively.

**Bioluminescence and fluorescence imaging**. All in vivo imaging experiments were performed using Xenogen IVIS Spectrum imaging system (Caliper Life Sciences). Prior to imaging, mice were injected intraperitoneally with 100 µl of phosphate-buffered saline containing D-Luciferin substrate (PerkinElmer). For biodistribution study, 100 µl of ZnS-QDH solution was injected into the tail vein of tumor bearing mice and the fluorescent imaging data were collected at 24 and 48 h after injection. After 48 h, animals were euthanized and selected major organs (liver, kidney, spleen, lung, and heart) were extracted for ex vivo imaging. Tumor growth was measured both by caliper measurement and bioluminescence imaging.

**Statistics**. We performed all statistical analyses with a one-way analysis of variance (ANOVA) test. Results are represented as mean ± sem. No animal or sample was excluded from the analysis.

**Data availability**. The authors declare that all the data supporting the findings of this study are available within the paper and its Supplementary Information files or from the author upon request.

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

## Acknowledgements

This research was supported by Canadian Institutes of Health Research (CIHR) operating grant and National Sciences and Engineering Research Council of Canada (NSERC) discovery grant. We thank Caifeng Wang and Tina Saberi Safaei for sharing their technical knowledge and advice on this work.

## Author contributions

L.Z., S.R.J., E.H.S. and S.O.K. formulated the project. P.M.A. performed the rheological tests, L.Z. and X.L. synthesized QDs, and F.F. assisted with the TEM imaging experiments. All of the in vivo studies were designed by L.Z., S.R.J. and S.A. and performed by S.A. L.Z., S.R.J. and X.L. designed and performed all remaining experiments. L.Z., S.R.J., E.H.S. and S.O.K. wrote and edited the manuscript.

## Additional information

**Competing interests:** The authors declare no competing financial interests.

