## [Peer Review File · Nature Communications]

Reviewers' comments:

Reviewer #1 (Remarks to the Author):

Review of "Multifunctional Quantum Dot-DNA Hydrogels" by Zhang and Jean, et. al.

The authors have addressed the majority of the previous comments from the previous submission and review with additional experiments and I believe that the manuscript could be published here after some minor corrections

1. Page 6, second paragraph, in which the swelling to initial concentration is discussed is very confusing and seems to be repetitive. Please clarify.
2. Page 11, second sentence, when you measure at 0, 24, and 48 hours the use of "every 24 hours up to 48 h" is a confusing way to state it, I assumed it was a typo and you were measuring much more often. Only by comparing to the graphs was it made clear.
3. Page 17-18, flow cytometry method. You do not clarify how you realize the QD alone experiments only the QDH. This is an important point that I would like to bring up in figure 4B. How do you know that you are delivering similar QD amounts if you don't know the number of QD per QDH?
4. In the figures (1 and TOC) I do not believe the label of "Cancer Cell" is appropriate the mechanism for uptake and release is the same whether it is a cancerous cell or not. You have made the targeted aspect clear in the labeling of the receptor and your differential uptake is not that good as to claim you only are delivering to the desired cells.
5. Something minor - in the legend of figure 5 it says unteated instead of untreated.
6. How are you calculating the QD QYs you report in table S1?
7. The data points in figure S5a,c are missing for the 3hr. Were they not taken or do they go above scale similar to the data in the main text?
8. In terms of citations – some references that are more focused on QDs and DNA should be included.
9. Finally I would reiterate that it would be nice to have an estimate of the number of QD per QDH. I am aware that the authors state that their previous attempts have been unsuccessful, but if you know the concentration of QDH (in the methods for flow cytometry and confocal microscopy you claim to add specific amounts) you could then digest them and use UV-Vis to determine a QD amount.

Reviewer #2 (Remarks to the Author):

This is a revised manuscript with additional data to support the stability and biocompatibility of QD hydrogel. Most of the concerns have been adequately addressed. One remaining question is how QD hydrogel nanoparticles were actually synthesized because according to Figure 3 b, QD hydrogel actually formed a gel network instead of discrete particles. It is also not clear what were the dose, dosing schedule, and route of administration used in the antitumor efficacy studies with regard to both Dox and QDs. The efficacy data is not that impressive given that many nanoparticles carrying Dox reported in the literature have shown significantly better antitumor activity than free Dox.

Reviewer #3 (Remarks to the Author):

I have had a chance to review a previous version of this manuscript, and raised the comments that control/comparison studies need to be carried to show whether the reported hybrid nanostructures indeed have important advantages. In the revised version, the authors have carried out these studies and have additional experimental data, which have addressed my previous concerns. Also, the authors appear to have satisfactorily addressed the criticisms raised by other referees. Thus, I would like to recommend this work for publication.

Response to reviewers' comments (comments in italics, our response in bold)

Reviewer #1:

Review of "Multifunctional Quantum Dot-DNA Hydrogels" by Zhang and Jean, et. al.

The authors have addressed the majority of the previous comments from the previous submission and review with additional experiments and I believe that the manuscript could be published here after some minor corrections

1. Page 6, second paragraph, in which the swelling to initial concentration is discussed is very confusing and seems to be repetitive. Please clarify.

We have revised this sentence in the manuscript to clarify the dependence of swelling capacity on the initial concentration of starting material.

2. Page 11, second sentence, when you measure at 0, 24, and 48 hours the use of "every 24 hours up to 48 h" is a confusing way to state it, I assumed it was a typo and you were measuring much more often. Only by comparing to the graphs was it made clear.

We agree with the reviewer and have changed the manuscript to clarify this sentence.

3. Page 17-18, flow cytometry method. You do not clarify how you realize the QD alone experiments only the QDH. This is an important point that I would like to bring up in figure 4B. How do you know that you are delivering similar QD amounts if you don't know the number of QD per QDH?

We have added details to the flow cytometry protocol for QD alone in the method section. We determined the amount of QD from the absorbance measurements of the QDs.

4. In the figures (1 and TOC) I do not believe the label of "Cancer Cell" is appropriate the mechanism for uptake and release is the same whether it is a cancerous cell or not. You have made the targeted aspect clear in the labeling of the receptor and your differential uptake is not that good as to claim you only are delivering to the desired cells.

We changed the "cancer cell" label to "cell" in Figure 1 and TOC.

5. Something minor - in the legend of figure 5 it says unteated instead of untreated.

We have corrected it to untreated.

6. How are you calculating the QD QYs you report in table S1?

The QY was calculated based on the reported standard method (Adv. Mater. 1997,

9, 230-232), which was performed using a Quanta-Phi integrating sphere coupled to a Horiba Fluorolog system. The excitation source is a monochromated Xe lamp. Following are the setting parameters used for the quantum efficiency measurements: an excitation wavelength of 440 nm, bandpass values of 2 and 2 nm for the excitation and emission slits, step increments of 1 nm, and integration time of 0.1 s per data point. Excitation and emission spectra were collected with the sample directly in the excitation beam path and with the sample offset from the beam path and the empty sphere. A calibrated neutral density filter with a known transmission was placed after the integrating sphere to measure the excitation intensity. The detector and integrating sphere were calibrated for spectral variance with a Newport white light source.

We have added this description into the method section.

7. The data points in figure S5a,c are missing for the 3hr. Were they not taken or do they go above scale similar to the data in the main text?

The results indeed go above scale similar to the data in the main text. We now indicate this in Figure S5a,c.

8. In terms of citations – some references that are more focused on QDs and DNA should be included.

We now include references that are more focused on QDs and DNA (refs 25, 26, 28, 29, and 37).

9. Finally I would reiterate that it would be nice to have an estimate of the number of QD per QDH. I am aware that the authors state that their previous attempts have been unsuccessful, but if you know the concentration of QDH (in the methods for flow cytometry and confocal microscopy you claim to add specific amounts) you could then digest them and use UV-Vis to determine a QD amount.

We thank the reviewer for the constructive comment. Using the calculation method suggested by the reviewer, we obtained the number of QDs to be approximately 178 ± 32 for the 80nm QDH.

Referee #2:

This is a revised manuscript with additional data to support the stability and biocompatibility of QD hydrogel. Most of the concerns have been adequately addressed. One remaining question is how QD hydrogel nanoparticles were actually synthesized because according to Figure 3 b, QD hydrogel actually formed a gel network instead of discrete particles. It is also not clear what were the dose, dosing schedule, and route of administration used in the antitumor efficacy studies with regard to both Dox and QDs. The efficacy data is not that impressive given that many nanoparticles carrying Dox

reported in the literature have shown significantly better antitumor activity than free Dox.

For the synthesis of QDH in Figure 3b, DNA-capped QDs (200 μ M) and Y-shaped DNA nanostructure (133 μ M) (ratio: 1.5:1) were mixed and incubated at 37°C for 2 hours.

The dosage of 4 mg/kg Dox-QDH and Dox was determined using equimolar amount of Dox. The amount of QDH (0.45 mg/kg) used was measured based on the weight of carrier used in Dox-QDH. The samples were administered into mice through tail-vein injection.

Referee #3:

I have had a chance to review a previous version of this manuscript, and raised the comments that control/comparison studies need to be carried to show whether the reported hybrid nanostructures indeed have important advantages. In the revised version, the authors have carried out these studies and have additional experimental data, which have addressed my previous concerns. Also, the authors appear to have satisfactorily addressed the criticisms raised by other referees. Thus, I would like to recommend this work for publication.

Thank you for your kind comments.

REVIEWERS' COMMENTS:

Reviewer #1 (Remarks to the Author):

The Authors have addressed all the points, the manuscript should be acceptable

Reviewer #2 (Remarks to the Author):

The authors have adequately addressed all concerns I have raised previously. It appears that critiques from other reviewers are all satisfactorily addressed. Recommend acceptance of this paper.